# Dengue pre-vaccination screening test evaluation for the use of dengue vaccine in an endemic area

**Umaporn Limothai**[1,2], **Sasipha Tachaboon**[1,2], **Janejira Dinhuzen**[1,2], **Taweewun Hunsawong**[3], **Prapapun Ong-ajchaowlerd**[3], **Butsaya Thaisomboonsuk**[3], **Stefan Fernandez**[3], **Supachoke Trongkamolchai**[4], **Mananya Wanpaisitkul**[4], **Chatchai Chulapornsiri**[4], **Anongrat Tiawilai**[5], **Thawat Tiawilai**[5], **Terapong Tantawichien**[6,7], **Usa Thisyakorn**[7], **Nattachai Srisawat**[1,2,7,8,9,10]*

1 Excellence Center for Critical Care Nephrology, King Chulalongkorn Memorial Hospital, Bangkok, Thailand, 2 Critical Care Nephrology Research Unit, Faculty of Medicine, Chulalongkorn University, Bangkok, Thailand, 3 Department of Virology, Armed Forces Research Institute of Medical Sciences, Bangkok, Thailand, 4 Department of Medicine, Banpong Hospital, Ratchaburi, Thailand, 5 Department of Medicine, Photharam Hospital, Ratchaburi, Thailand, 6 Division of Infectious Diseases, Department of Medicine, Faculty of Medicine, Chulalongkorn University, Bangkok, Thailand, 7 Tropical Medicine Cluster, Chulalongkorn University, Bangkok, Thailand, 8 Division of Nephrology, Department of Medicine, Faculty of Medicine, King Chulalongkorn Memorial Hospital, Bangkok, Thailand, 9 Center for Critical Care Nephrology, The CRISMA Center, Department of Critical Care Medicine, University of Pittsburgh, School of Medicine, Pittsburgh, PA, United States of America, 10 Academy of Science, Royal Society of Thailand, Bangkok, Thailand

* drnattachai@yahoo.com

**Data Availability Statement:** All relevant data are within the manuscript and its Supporting Information files.

## Abstract

### Background

The dengue vaccine (Dengvaxia) is only recommended for individuals with prior dengue infection (PDI). This study aimed to perform a serosurvey to inform decision-making for vaccine introduction and identify appropriate target populations. We also evaluated the performance of the serological tests using plaque reduction neutralization test (PRNT) as a reference test in identifying PDI to determine suitability for pre-vaccination screening.

### Methods

We enrolled 115 healthy individuals between 10 and 22 years of age living in the Ratchaburi province of Thailand. The serum samples were tested by PRNT to measure the prevalence and concentration of serotype-specific neutralizing antibodies. The performance of the IgG rapid diagnostic test (RDT, SD Bioline, Korea) and IgG enzyme-linked immunosorbent assay (ELISA, EUROIMMUN, Germany) in identifying PDI were evaluated by using PRNT as a reference method.

### Results

Ninety-four (81.7%) individuals neutralized one or more dengue serotypes at a titer threshold greater than or equal to 10. Multitypic profiles were observed in 70.4% of the samples which increased to 91.9% in subjects aged 19–22. Among monotypic samples, the highest

**Funding:** This research project is supported by Ratchadapiseksompotch Fund Chulalongkorn University and the Second Century Fund (C2F), Chulalongkorn University (Umaporn Limothai received the grant). Nattachai Srisawat received funding from the Jongkolneenithi foundation, the Medical Association of Thailand, and the Tropical Medicine Cluster, Chulalongkorn University. The funders had no role in study design, data collection, and analysis, decision to publish, or preparation of the manuscript.

**Competing interests:** The authors have declared that no competing interests exist.

proportion was reactive against DENV-1 followed by DENV-2, DENV-3, and DENV-4. The highest anti-dengue antibody titers were recorded against DENV-1 and increased with age to a geometric mean NT50 titer (GMT) of 188.6 in the 19–22 age group. While both RDT and ELISA exhibited 100% specificity, RDT demonstrated low sensitivity (35%) with ELISA displaying much greater sensitivity (87%).

## Conclusions

Almost 80% of adolescents and youth in Ratchaburi province had already been exposed to one or more of the dengue virus serotypes. The dengue IgG RDT displayed low sensitivity and is likely not be suitable for dengue pre-vaccination screening. These results support the use of IgG ELISA test for dengue vaccination in endemic areas.

## Introduction

Dengue is a mosquito-borne viral disease that occurs across tropical and subtropical regions. Dengue virus consists of four serotypes with antigenic differences (DENV-1 to 4) [1]. It has a broad clinical spectrum that includes both severe and non-severe clinical manifestations [2]. The first dengue vaccine, a live attenuated tetravalent dengue vaccine (Dengvaxia or CYD-TDV), was developed by Sanofi Pasteur and was licensed and available in select countries. The efficacy trials showed that Dengvaxia protected against severe dengue in people who had exposure to dengue prior to vaccination [3–5]. The trials also showed evidence of an increased long-term risk of severe dengue in vaccinated persons who had not been exposed to dengue [5]. The World Health Organization (WHO) Strategic Advisory Group of Experts (SAGE) currently recommends only using Dengvaxia in children 9–16 years old living in highly dengue-endemic areas [6]. When selecting areas and populations to be targeted for vaccination, prior dengue infection (PDI) as measured by seroprevalence should be at least 80% at age 9 [7]. To minimize potentially harmful effects, we should ensure that the age group targeted for vaccination has sufficient pre-existing dengue immunity to benefit from vaccination [7].

Age-stratified serosurveys are currently the best method for selecting populations appropriate for vaccination. Immunoglobulin G (IgG) class antibodies are useful serological markers for PDI as they can persist over an individual's lifetime [8]. Plaque reduction neutralization test (PRNT) is the most specific serological test for determining type-specific neutralizing antibodies to an infecting virus. This technique can confirm the presence of neutralization antibodies against Flaviviruses in serum samples [9, 10]. However, PRNT is a labor-intensive, relatively costly test making it inconvenient for routine surveillance purposes [10, 11]. Commercial IgG rapid diagnostic test (RDT) and IgG enzyme-linked immunosorbent assay (ELISA) are commonly used to diagnose acute dengue infection [12]. Performance data from tests identifying PDI remains limited [13]. A recent systematic review of RDT showed sensitivities and specificities between 80% and 100% for dengue IgG detection compared to ELISA [13]. These studies evaluated IgG detection in early convalescent samples and those with presumed acute primary or secondary infection. No study evaluated the performance of RDT in samples with previous DENV infection [13].

This study performed a comparative serosurvey to support decision-making for dengue vaccine introduction at nine years. We also evaluated the performance of the serological tests as a pre-screening tool in identifying PDI in order to determine their suitability for pre-vaccination screening in Thailand.

## Materials and methods

### Ethics statement

The study was conducted according to the Helsinki Declaration and Good Clinical Practice guidelines. The study protocol was approved by the Ethics Committee of Banpong Hospital (REC 008/2562) and Potharam Hospital (REC 31/2562). Written informed consent was obtained from subjects ≥18 years old and parents of subjects <18 years old.

### Cells and viruses

The PRNT assays were completed in LLC-MK2 (Rhesus monkey kidney cells) obtained from ATCC. The cells were grown in M-199 supplement with 20% heated-inactivated fetal bovine serum (HIFBS, Invitrogen, US), 1× L-glutamine (Invitrogen, US) and 100 units of penicillin-streptomycin (P&S, Invitrogen, US) at 35˚C with 5% $CO_2$ incubator. A concentration of cells at $4 \times 10^4$ per ml in M-199 with 10% HIFBS, 1× L-glutamine, 100 units of P&S was prepared and added 1.5 ml per well in 12-well plate which continued being cultured at 35˚C with 5% $CO_2$ for 3 days before using in the PRNT assay. The four serotypes of DENV including DENV-1 (16007-Passage SM-2 C6/36-5), DENV-2 (16681-Passage MIK2-3, C6/36-1, SM-2, C6/3), DENV-3 (16562-Passage MK2-3, C6/36-2, SM-1, C6/36-6) and DENV-4 (C0036/06-Passage C6/36-8) were used in the PRNT assay. Viruses were propagated in a C6/36 mosquito cell line to obtain a sufficient titer (700PFU/ml) for testing. Cytopathic effects (CPEs) observation was completed daily to ensure that the virus was harvested at the appropriate time. Virus titers were measured by plaque titration assay. Viruses were aliquot in small volume and stored at -70˚C until used without repeat freeze-thawing. We titered the virus after the first freeze using a procedure similar to PRNT assay to determine virus titer and also for virus dilution to reach 70 PFU/100 ul.

### Study location and population

The study was conducted in Banpong and Potharam district, Ratchaburi province, an urban center located in the western part of Thailand. Healthy children between 9 and 22 years at the time of enrollment and residing in Banpong or Potharam district were eligible to participate. Healthy children were defined as those in good health based on medical history and physical examination. We excluded children with acute febrile illness (body temperature > 37.5˚C) or moderate to severe acute illness/infection on the day of enrollment. Using the sample size calculation by Malhotra RK. and Indrayan A [14]. and assuming 91% seroprevalence [15, 16], expected sensitivity and specificity of 95% and desired precision of 0.05, we estimated a need for a minimum of 80 samples. Including indeterminate and invalid results, we tested 115 samples. To examine the DENV seroprevalence in different age groups, we classified participant age into three categories: 9–14 years, 15–18 years, and 19–22 years. Specimens were collected from August 2019 to August 2020. The blood samples were collected in anticoagulant-free vacutainer tubes. The sera were separated and maintained in refrigeration at 2–8˚C. The specimens were then transported on the day of collection to the central laboratory at Chulalongkorn hospital and stored at -80˚C before testing.

### Plaque Reduction Neutralization Test (PRNT)

PRNT was performed to determine the level of virus type-specific neutralizing (NT) antibodies using standard methods, as previously described [9, 17, 18]. The serum was heat-inactivated at 56˚C for 30 minutes. In an ice bath, a 0.3 ml/tube of each 4-fold dilutions of the test sera beginning with a 1:10 dilution (1:10, 1:40, 1:160, 1:640 and 1:2,560) including a 0.3 ml serum diluent

tube as a virus control well (MEM with 10% HIFBS) was mixed with an equal volume of each reference virus (DENV serotype noted above). The virus-serum mixture was then incubated in a 35°C water bath for 1 hour. The virus-serum mixture was then inoculated (0.1 ml per well) into the 3-day old LLC-MK2 cells in duplicate wells of a 12-well plate and incubated at room temperature for 1 hour on a rocker platform. After removal of excess virus-serum mixture, the first overlay medium containing 0.9% low-melting point agarose (LMP Ultra PureTM LMP agarose, Invitrogen, USA), in-house Hank's BSS, 1X vitamins (Invitrogen, US), 1X amino acids (Invitrogen, US), 5% HIFBS, L-glutamine, and 7.5% sodium bicarbonate was added (1 ml per well) and allowed to solidify for 15 minutes at room temperature. Plates were incubated at 35°C in a 5% $CO_2$ incubator for 4–6 days. At the end of the incubation period, stained cells with a second overlay medium (0.9% LMP, in-house Hank's BSS, 1X vitamins, and 1X amino acids) containing 4% neutral red (Sigma, US) 1 ml per well were allowed to solidify for 15 minutes. Plates were incubated overnight at 35°C in a 5% $CO_2$ incubator before manually counting the number of plaques. Positive control wells (virus without sera) were established for each assay run to ensure infectivity of the cell monolayer. Formation of 25–50 plaques/well was considered acceptable. Positive serum control was pooled high-titered convalescent sera against four dengue serotypes diluted with pooled normal human sera to reach the appropriate NT50 endpoint and aliquoted in small volume and frozen for use.

Plaque reduction neutralization endpoints were calculated using probit analysis. The percentage of plaques counted in test sera was compared with the number of plaques from the control preparation. Log dilution of test sera preparation (i.e., 1:10, 1:40, 1:160, 1:640, 1:2,560) were plotted along the X-axis with percent reduction in plaque count on the Y-axis. Percent count plaque reductions for each test sample at each dilution was plotted and a best-fit line was derived. The reciprocal of the lowest dilution of test sera to neutralize 50% of the control virus input represents the NT50 titer. Therefore, the NT50 titer is calculated by counting plaques and reporting the titer as the reciprocal of the last serum dilution to show a 50% reduction of the control plaque count based on the back-titration of control plaques. In this study, NT50 titers≥10 was considered positive for virus-neutralizing activity. An individual who had antibody titers <10 for the four serotypes was classified as naïve, while an individual with NT50 antibody titers ≥10 in only one serotype (no NT titer to other serotypes) was classified as monotypic. Lastly, an individual with NT50 antibody titers ≥10 for more than one serotype was classified as multitypic.

## Dengue duo rapid test kit

Dengue nonstructural protein 1 (NS1) antigen and dengue IgG/IgM antibody were examined at the enrollment using SD Bioline Dengue Duo NS1 Ag & IgG/IgM (catalog number 11FK46, SD Bioline, Korea) according to manufacturer instructions. The kit contained two test devices. The left side device is for the dengue NS1 antigen test, while the right side is for the dengue IgG/IgM test. The serum specimen was thawed to room temperature (15°C—30°C) before use. For the NS1 antigen, 100 μL of undiluted serum sample was added to the left sample well and incubated at room temperature. The results were then read within 15–20 minutes. The presence of two colored lines including control line (C) and test line (T) in the result window indicates that the specimen is positive for the NS1 antigen, while the presence of only the C line indicates that the NS1 antigen is not present in the specimen or is present below the detectable levels.

For IgG/IgM, 10 μL of undiluted serum sample was added to the right sample well. This was followed by adding four drops (90–120 μL) of assay diluent to the round assay diluent well and incubated at room temperature. The results were read after 15–20 minutes. The presence

of two colored lines (C and M) in the result window indicates that the specimen is positive for IgM antibodies to dengue. The presence of two colored lines (C and G) indicates that the specimen is positive for IgG antibodies to dengue. In addition, the presence of three colored lines (C, M, and G) indicates that the specimen is positive for both IgM and IgG antibodies to dengue. In contrast, the presence of only the C line indicates that IgG and IgM antibodies to dengue are not present in the specimen or are present below the detectable levels [19]. The test results were investigated and interpreted by three different readers to minimize bias.

## Dengue IgG ELISA

Anti-dengue IgG antibodies were assessed using commercial enzyme-linked immunosorbent assays (catalog number EI266b-9601G, EUROIMMUN, Lübeck, Germany) according to manufacturer instructions. Serum samples were diluted at 1:101 and 100 μL of the mixture was then added to the 96-well microplates coated with purified DENV-2 particles. Due to the high structural similarity of DENV-1-4, the use of a single virus type is sufficient for the reliable detection of antibodies of all four virus types. After incubating for 60 minutes at 37°C, the wells were washed three times using 300 μl of working strength wash buffer for each wash and 100 μL of enzyme conjugate (peroxidase-labeled anti-human IgG) was added to each well. After incubating for 30 minutes at room temperature, the wells were washed three times using 300 μl of working strength wash buffer for each wash, and the conjugated complex was visualized by adding 100 μL of chromogen/substrate solution into each of the microplate wells. After incubating for 15 minutes at room temperature (protected from direct sunlight), the reaction was stopped by adding 100 μl of stop solution into each microplate well. Photometric measurement of the color intensity was made at a wavelength of 450 nm and a reference wavelength between 620 nm and 650 nm within 30 minutes of adding the stop solution. The IgG results were reported quantitatively according to manufacturer instructions using the standard curve obtained by point-to-point plotting of the extinction values measured for the three calibration sera against the corresponding units (linear/linear). A specimen was considered IgG positive if the result was ≥22 relative units (RU)/ml, borderline if ≥16 and <22 RU/ml, and negative if <16 RU/ml.

## Statistical analysis

Descriptive statistics for seropositive rate and other characteristics were computed. Continuous variables are presented as the mean ± one standard deviation (SD) in cases of a normal distribution and as a median and interquartile range (IQR) in cases of non-normal distribution. The comparison was performed by one-way ANOVA or Kruskal–Wallis test as appropriate. Categorical variables were presented as numbers with percentages and compared using Chi-square test or Fisher's exact test as appropriate. P-value <0.05 was considered significant. The geometric mean titer and the 95% CI for each age group and dengue serotype were calculated for all samples based on their DENV PRNT results. To calculate the GMT, samples with an NT50 titer <10 were given the value of 5. All analyses were performed using IBM SPSS Statistics software version 22.0 (IBM Corp., Armonk, NY) and figures were drawn using GraphPad Prism 8 (GraphPad Software Inc., California, USA).

## Disclaimer

Material has been reviewed by the Walter Reed Army Institute of Research. There is no objection to its presentation and/or publication. The opinions or assertions contained herein are the private views of the author and are not to be construed as official, or as reflecting the true views of the U.S. Department of the Army or the U.S. Department of Defense.

**Table 1. Demographic data.**

| Characteristics | Total (N = 115) |
|---|---|
| Gender, male, (N, %) | 67 (58.3) |
| Age, years (median, range) | 16 (10–22) |
| 10–14 (N,%) | 38 (33.0) |
| 15–18 (N,%) | 40 (34.8) |
| 19–22 (N,%) | 37 (32.2) |
| Area | |
| Ban Pong (N,%) | 61 (53.0) |
| Potharam (N,%) | 54 (47.0) |

## Results

### Demographic data

Table 1 provides general characteristics of the 115 participants included in the study. Their age ranged from 10 to 22 years, with a median of 16 years (IQR 13–20). More than half (67, 58.3%) were males. The participants were predominantly (53%) from Ban Pong District.

### Seroprevalence of DENV among participants

Serological evidence of previous exposure to DENV as denoted by NT50 titers $\geq$ 10 was identified in the serum samples of 94 (81.7%) participants. Seroprevalence did not significantly differ by gender ($p$ = 0.113). To examine the DENV seroprevalence in different age groups, we classified participants into three categories: 10–14 years, 15–18 years, and 19–22 years. The data showed that older age groups were associated with increased seropositivity (p = 0.005). We also examined the seroprevalence by IgG ELISA and IgG rapid test. The results revealed a decreased overall seropositive rate of 71.3% by IgG ELISA and 28.7% by IgG rapid test (Fig 1). WHO recommends the use of NT90 instead of NT50 in endemic areas to decrease the background serum cross-neutralization among flaviviruses [20]. We performed additional PRNT data analysis using a higher cut-off and found a decreased overall seropositive rate of 68.7% by NT90 compared to our original 81.7% by NT50 (S1 Table).

Based on the RDT, we found no positive NS1 results which indicated no early dengue infection case in this cohort. There were 2 IgM positive cases reflecting the asymptomatic acute

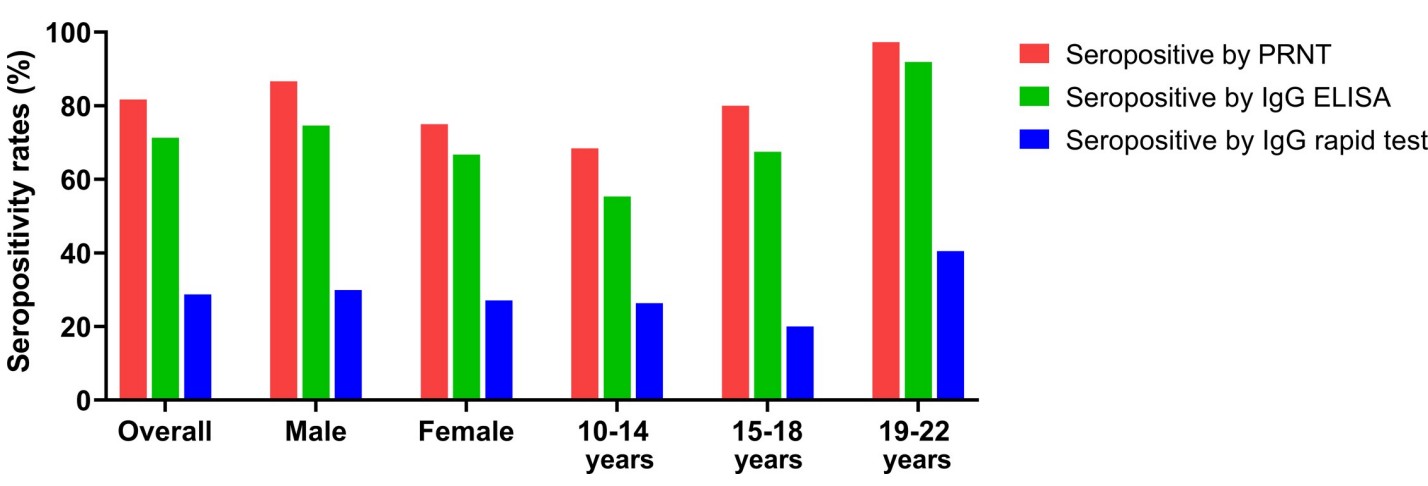

**Fig 1. Seroprevalence of DENV by PRNT, IgG ELISA, and IgG rapid test.**

primary dengue infection and 3 IgM and IgG positive cases suggesting a late primary or early secondary dengue infection.

### Neutralizing antibody profile distribution stratified by age

Samples were categorized according to the NT50 profile. Multitypic profiles were observed in 70.4% of the subjects with 55.3% among 10–14 years old, 65.0% in the 15–18 years old, and 91.9% in the 19–22 years old (Fig 2).

There were 31.6% naïve subjects in the 10–14 years old group, 20.0% of the 15–18 years old group and 2.7% of the 19–22 years old groups. The overall sample had a rate of 18.3% with no detectable neutralizing dengue antibodies (NT50 titer<10, Fig 2).

Among monotypic samples, the highest proportion were reactive against DENV-2 followed by DENV-1, DENV-3, and DENV-4, a trend observed in the 19–22 age group. The proportion

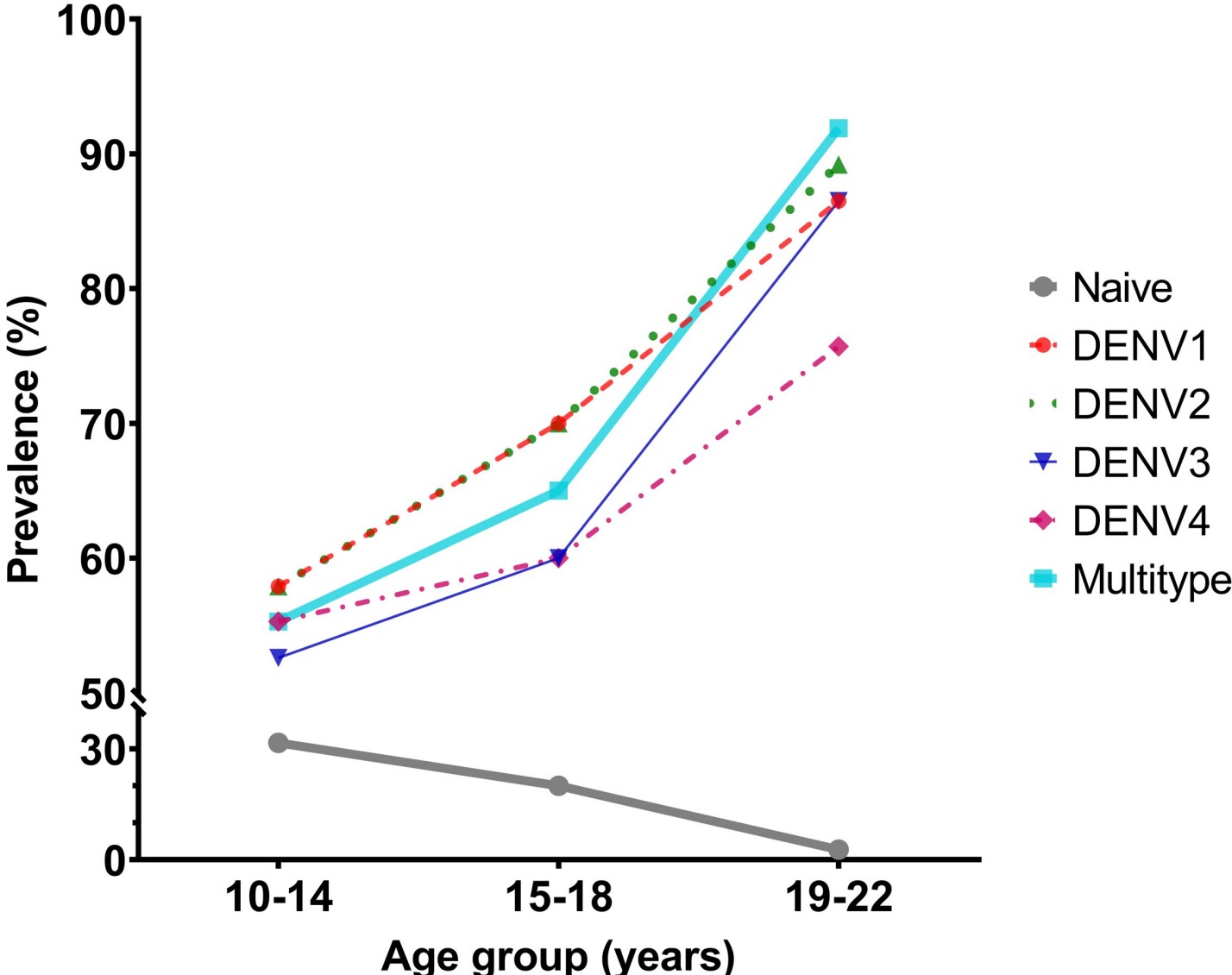

**Fig 2. Prevalence of naïve (NT50 titer<10), monotypic (NT50 titer≥10 against to only one dengue serotype) or multitypic (NT50 titer≥10 against more than one serotype) neutralizing antibody profile by age group.**

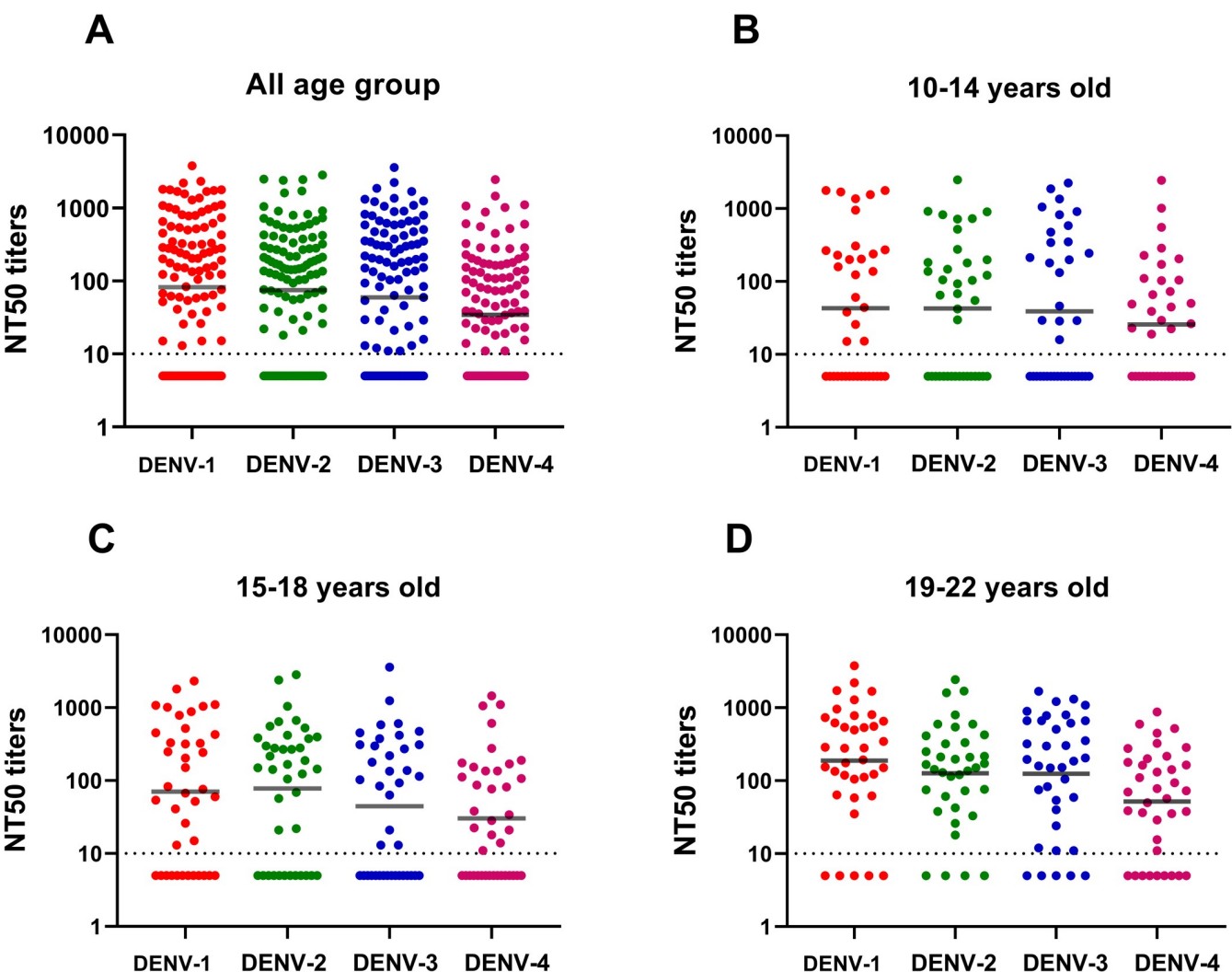

**Fig 3. Dengue serotype-specific geometric mean NT50 titer (GMT) by age-group.** The long solid line indicates the threshold for seropositivity (an NT50 titer ≥10).

of samples that were reactive against DENV-1 and DENV-2 were evenly distributed among the 10–14 and 15–18 age groups.

The distribution of the geometric mean of NT50 titer (GMT) among the entire sample is shown in Fig 3A. DENV-1 had the highest overall GMT (82.2) followed by DENV-2 (74.5), DENV-3 (59.3) and DENV-4 (34.2). The trends were also observed in the 10–14 age group (Fig 3B) with GMT by DENV-1-4 of 43.1, 42.8, 38.8, and 25.8, respectively, and for the 19–22 age group (Fig 3D) with 188.6, 126.0, 124.9 and 51.7, respectively. For the 15–18 age group (Fig 3C), DENV-2 had the highest GMT (77.7) followed by DENV-1 (70.5), DENV-3 (44.6) and DENV-4 (30.4).

## Performance of the dengue IgG rapid test and dengue IgG ELISA for determination of dengue serostatus

We assessed the performance of the IgG rapid test and IgG ELISA for detection of dengue pre-exposure using the PRNT as the reference assay. Of the 94 patients classified as seropositive by NT50 (titers ≥ 10), 61 (64.9%) tested negative by dengue IgG rapid test and 12 (12.8%) tested

**Table 2. Performance of the dengue IgG rapid test and dengue IgG ELISA for serostatus determination using the plaque reduction neutralization test (PRNT) as the reference assay.**

| Test | Dengue rapid test (IgG) | | Dengue ELISA (IgG) | |
|---|---|---|---|---|
| | Positive | Negative | Positive | Negative |
| Positive PRNT (n,%) | 33 (35.1%) | 61 (64.9%) | 82 (87.2%) | 12 (12.8%) |
| Negative PRNT (n,%) | 0 (0.0%) | 21 (100.0%) | 0 (0.0) | 21 (100.0%) |
| Sensitivity | 35.1% | | 87.2% | |
| Specificity | 100.0% | | 100.0% | |
| Positive predictive value | 100.0% | | 100.0% | |
| Negative predictive value | 25.6% | | 63.6% | |

negative by dengue IgG ELISA. For the dengue IgG rapid test, sensitivity was 35.1% and specificity was 100%. When we evaluated the dengue IgG ELISA to detect prior dengue infection, specificity was unchanged but sensitivity increased to 87.2% compared to IgG rapid test. Positive predictive values (PPV) for both tests were 100%. Negative predictive values (NPV) were 25.6% and 63.6% for dengue rapid test and IgG ELISA, respectively (Table 2).

The relationship between NT50 titer and dengue IgG level (relative RU/mL by ELISA) was also examined. Results showed that NT50 titer had a positive correlation with IgG level (r = 0.701, p<0.001 for DENV-1, r = 0.771, p<0.001 for DENV-2, r = 0.667, p<0.001 for DENV-3, and r = 0.725, p<0.001 for DENV-4). The distribution of dengue IgG level and NT50 titer by age is demonstrated in Fig 4.

## Discussion

We conducted a dengue seroprevalence study which identified serological evidence for the circulation of all four dengue serotypes among adolescents and youth in Ratchaburi province, Thailand. This province is located approximately 100 km west of Bangkok. It lies between the Maeklong River on the east and the Thai-Myanmar border on the west. We have a collaborative research network in this province. A previous dengue disease surveillance study on

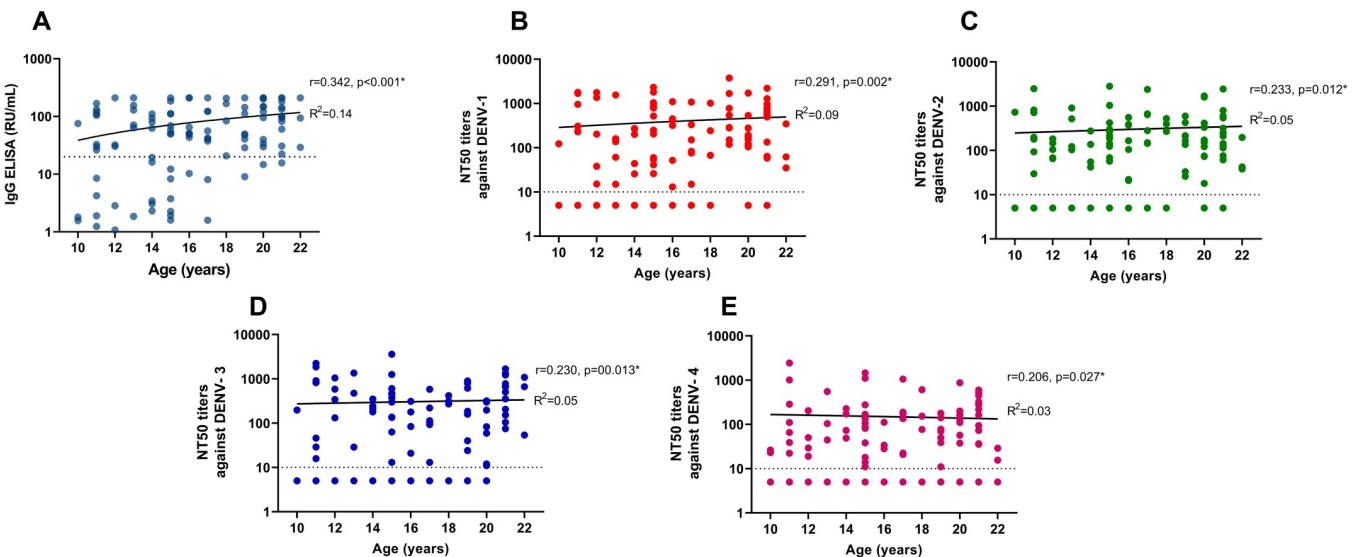

**Fig 4. Distribution of dengue IgG level (relative units per milliliter by ELISA) and NT50 titer by age.** The dashed line indicates the threshold for dengue IgG ELISA (20 RU/ml) and NT50 (a titer of 10).

approximately 3000 primary-school children in seven schools in Muang district of Ratchaburi province indicated that approximately 4% of children had laboratory-confirmed dengue annually. All four dengue virus types were also found to be the cause of illness in children in all seven schools. The study provided evidence that the Muang district of Ratchaburi province could be a suitable area for dengue vaccine testing [21].

Our study generated data on serotype-specific prevalence in areas where little data were previously available. The results indicated that approximately 82% of individuals neutralized one or more dengue serotypes shown by PRNT (NT50 titers $\geq$ 10). The proportion of individuals with exposure to more than one serotype was found to increase with age. The predominant serotypes were DEN-1 and DEN-2 in the 10–14 and 15–18 age groups born during 2003–2011. Similar to previous reports, DENV-1 and DENV-2 were the most commonly reported serotypes in Thailand's national and/or regional studies between 2000 and 2010 [22]. DEN-2 was found predominant in the 19–22 years old age groups born during 1999–2002. This result was consistent with a previous surveillance study in the Ratchaburi province which indicated that the predominant serotypes were DEN-2 in 1999 and 2002 [23].

The seroprevalence data previously found among Thai individuals living in central and southern Thailand (ages 6 months to 60 years) indicated that the overall seropositive rate for anti-DENV IgG antibodies by ELISA (catalog numbers EI266b-9601G, EUROIMMUN, Lübeck, Germany) was 79%, which was similar to this study [15]. More recently, a higher seroprevalence rate was observed in northeastern Thailand (approximately 90%, ages 5 to 93 years) by the same ELISA kit. The seroprevalence was 100% for participants over the age of 25 years [16]. A previous study of healthy children in Bangkok indicated that 9%-23% of infants aged 9–18 months had neutralizing antibodies against one or more DENV serotypes. It is thought that antibodies were maternally transferred. DENV-1 was the most prevalent followed by DENV-3, DENV-2, and DENV-4 [24]. A recent study has investigated dengue antibodies seroprevalence among mothers living in Ban Pong, Ratchaburi Province. The results showed that 97% of mothers had dengue neutralizing antibodies for at least one dengue serotype in their sera. Moreover, the proportion of cord sera with dengue neutralizing antibodies from all four dengue serotypes was high at birth and similar to their mother's sera [25]. Based on these results, the antibodies in infants may have been of maternal origin. In older children, the antibodies are thought to be the result of exposure to infected mosquitoes.

As there is an increased risk of hospitalization and severe dengue in seronegative individuals, the WHO SAGE committee recommended using Dengvaxia only in regions with seroprevalence of 80% or more at the age of 9 years [7]. Our data found that the seropositive rate among 10–14 age groups was 68%. This indicated that such vaccination programs might not be safe and effective in such a young age population. When tetravalent dengue vaccines become available for use in dengue-endemic countries, it should be carefully considered whether to target children who are 14 years or younger.

This study also shows that both RDT and ELISA have very high specificity in detecting IgG for identifying PDI. However, the RDT for detecting IgG demonstrated low sensitivity (35%), while the ELISA displayed high sensitivity (87%). In addition, the PPV for both tests was 100% indicating that the probability that a person with a positive test will be truly seropositive was 100%. Both RDT and ELISA could be used to assess safety from the adverse effects of the vaccination. In contrast, the NPV was 25% and 63% for the RDT and ELISA. The probability that an individual with a negative RDT test will be truly seronegative was only 25% leaving a 75% chance of seropositivity. If we select the RDT to identify PDI, persons with a negative RDT test result will have a 75% chance of missing the opportunity for vaccine protection.

According to WHO recommendations [7], the screening tests should have a high specificity to minimize individual risk and the inadvertent use of the vaccine in seronegative persons who

have false-positive test results. The test should also have high sensitivity to maximize individual and population benefit by identifying a high proportion of previously exposed persons who can benefit from vaccination. The Global Dengue and Aedes-transmitted diseases Consortium (GDAC) suggested that any decision about implementing a prevaccination screening strategy with the commercially available tests at the country level requires cautious evaluation. In a high seroprevalence area, a test with very high sensitivity is needed. In a low seroprevalence area, a very high specificity test is crucial; however, a national implementation may not be cost-effective [26].

Both RDTs and ELISA are widely used for the diagnosis of DENV infection [11]. Dengue RDTs are convenient for surveillance in resource-limited settings since they require less essential laboratory equipment. Currently, no study has evaluated the performance of RDT (SD Bioline) for previous DENV infection. A recent systematic review indicated that the sensitivity and specificity of dengue IgG RDTs were typically above 75% and 80%, respectively, compared to IgG ELISA testing in acute secondary DENV infection and convalescent time points after recent illness [13].

Bonaparte M. et al. evaluated the performance of RDT (Biocan Diagnostics Inc) and ELISA (SciMedx) in identifying individuals with PDI in Puerto Rico using PRNT as the reference test. Both RDT and ELISA showed high specificity (>99%) but moderate sensitivity (61% and 76%, respectively) [27]. In our evaluation, the RDT showed a disappointing sensitivity compared to ELISA. Based on results from the present study, the use of RDT to screen for PDI would miss 65% of persons who are seropositive and classified in the seronegative group thereby missing an opportunity for dengue vaccination. This suggests that the dengue IgG RDT (SD Bioline) may not appropriate for identifying PDI, especially in endemic areas. The desired characteristics of dengue pre-vaccination screening tests have been debated [28, 29]. It has recently been recommended that both sensitivity and specificity be at least 90% [26]. A newly developed, automated immunoglobulin fluorescence immunoassay for determining dengue serostatus has been recently developed. However, its specificity and sensitivity are still insufficient for pre-vaccination screening [30].

IgG ELISA measures total anti-DENV binding antibodies, whereas PRNT measures the neutralization capacity of the antibody response which would be a part of the binding antibody and more specific to a particular virus than ELISA [9]. The IgG ELISA might be more sensitive and less specific as it also detects non-neutralizing and cross-reactive antibodies to other flaviviruses within a serum specimen [31]. We found that persons who tested positive for dengue by IgG ELISA also tested positive using PRNT. This would minimize the likelihood of false positives in settings with past or current co-circulation of other flaviviruses. In this study, we used the ELISA index cutoff values for seropositivity as stated in the product insert. However, it was unclear whether the ELISA cutoff values recommended by the manufacturer were suitable for prevaccination screening. Doum D. et al. reported that the IgG ELISA optical densities (OD) with age were consistently higher in rural settings ($R^2$ = 0.46–0.54) and lower in urban settings ($R^2$ = 0.15–0.28) [16]. Our urban study indicated that the IgG ELISA titers (RU/ml) were low ($R^2$ = 0.14). We also found a positive correlation between IgG ELISA titers and NT50 titers.

We identified several limitations in our study. First, the sample collection was restricted to the Ratchaburi province which may not reflect the entire Thai country. Second, children under ten years old were not available for this study, so we could not determine the seroprevalence among 9-year-old children which is also a targeted age for vaccination. Third, we tested only one RDT (SD Bioline) and one ELISA kit (EUROIMMUN). It should be noted that the sensitivity for identifying PDI may vary among different companies. It is important to realize that cross-reactivity of anti-DENV antibodies with other flaviviruses could not be excluded by

most commercial antibody-based assays. Dengue and Japanese encephalitis virus (JEV) are in the same flavivirus family which share common antigenic epitopes and where high cross-reactivities are not uncommon. In Thailand, a mouse brain-derived-inactivated JEV vaccine was initially included in the routine immunization schedule in 1990. Since 2000, it has been part of the vaccination schedule for all Thai children at 12–18 months of age [32]. For this reason, all participants in this study were likely vaccinated against JEV for at least nine years. However, the detailed vaccination history of all participants was not collected. Cases of JEV encephalitis have been reported even among vaccinated persons, so natural JEV infection cannot be ruled out [33]. Convalescent anti-dengue antibody responses after dengue infection in JEV vaccinated and other individuals who lived in JEV endemic areas were dengue-specific and higher than anti-JEV as determined by PRNT [34]. A previous study found that the sera of patients with JEV showed cross-reactivity to DENV IgM and/or IgG ELISA, but the cross-reactivity was not detected in the neutralization tests against DENV [35]. In addition, Saito Y. et al. reported cross-reactivity to DENV in an anti-DENV IgG ELISA after JEV vaccination. However, the DENV neutralizing antibodies were not detected in post-JEV immunization samples indicating that neutralization tests are more specific than ELISA [36]. The serological data obtained from this study should be interpreted with caution since the samples were obtained in Thailand where several known flaviviruses have been reported.

Despite these limitations, this study provides the most recent seroprevalence data for dengue viruses from the western part of Thailand. We performed an age-stratified serosurvey using PRNT to distinguish specific DENV serotypes and estimate the true incidence of immunity to DENV. The study also produced data on RDT and ELISA performance in identifying PDI which can be used to determine suitability for pre-vaccination screening.

## Conclusion

This study confirms the distribution of multiple dengue serotypes in Thailand. The frequency of DENV infection increased in a time-dependent manner among children and young adults as the higher prevalence of multitypic DENV serotypes was observed in the older age group. Due to the lower sensitivity of RDT for identifying PDI, those with a negative RDT test result might lose the benefits of vaccination. Therefore, we do not recommend using RDT to identify PDI in this specific setting. Anti-dengue IgG ELISA is more suitable than RTD for vaccine introduction and identifying which populations to target.

## Supporting information

**S1 Appendix.**
(XLSX)

**S2 Appendix. STROBE statement—checklist of items that should be included in reports of observational studies.**
(DOC)

**S1 Table. Seroprevalence of DENV by PRNT (NT50, NT80 and NT90), IgG ELISA, and IgG rapid test.**
(DOCX)

## Acknowledgments

We are grateful to the volunteer children and their parents for participating in the study. We thank the medical staff at Banpong hospital and Potharam hospital for their support of this

project. We also thank Stefan Fernandez and the laboratory staff of the Department of Virology, Armed Forces Research Institute of Medical Sciences (Bangkok, Thailand) for PRNT assay training and reagents.

## Author Contributions

**Conceptualization:** Usa Thisyakorn, Nattachai Srisawat.

**Data curation:** Sasipha Tachaboon, Janejira Dinhuzen.

**Formal analysis:** Umaporn Limothai.

**Funding acquisition:** Terapong Tantawichien, Nattachai Srisawat.

**Investigation:** Nattachai Srisawat.

**Methodology:** Umaporn Limothai, Sasipha Tachaboon, Janejira Dinhuzen.

**Resources:** Supachoke Trongkamolchai, Mananya Wanpaisitkul, Chatchai Chulapornsiri, Anongrat Tiawilai, Thawat Tiawilai, Nattachai Srisawat.

**Supervision:** Terapong Tantawichien, Usa Thisyakorn, Nattachai Srisawat.

**Visualization:** Umaporn Limothai, Nattachai Srisawat.

**Writing – original draft:** Umaporn Limothai.

**Writing – review & editing:** Umaporn Limothai, Taweewun Hunsawong, Prapapun Ongajchaowlerd, Butsaya Thaisomboonsuk, Stefan Fernandez, Usa Thisyakorn, Nattachai Srisawat.

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
