## [Decision Letter · Decision Letter 0]

4 Jun 2021

PONE-D-21-09490

Dengue pre-vaccination screening test evaluation for the use of dengue vaccine in an endemic area

PLOS ONE

Dear Dr. Srisawat,

Thank you for submitting your manuscript to PLOS ONE. After careful consideration, we feel that it has merit but does not fully meet PLOS ONE’s publication criteria as it currently stands. Therefore, we invite you to submit a revised version of the manuscript that addresses the points raised below during the review process.

We look forward to receiving your revised manuscript.

Kind regards,

Ray Borrow, Ph.D., FRCPath

Academic Editor

PLOS ONE

Journal Requirements:

Reviewers' comments:

Reviewer's Responses to Questions

**Comments to the Author**

1. Is the manuscript technically sound, and do the data support the conclusions?

Reviewer #1: Partly

Reviewer #2: No

2. Has the statistical analysis been performed appropriately and rigorously? 

Reviewer #1: Yes

Reviewer #2: No

3. Have the authors made all data underlying the findings in their manuscript fully available?

Reviewer #1: Yes

Reviewer #2: No

4. Is the manuscript presented in an intelligible fashion and written in standard English?

Reviewer #1: Yes

Reviewer #2: Yes

5. Review Comments to the Author

Reviewer #1: Major issues:

A major part of flavivirus serology is unfortunately exclusion diagnostic. Therefore informative specificity analyses should include non-DENV samples, especially in a JEV endemic country as JEV is know to cause cross-reactivity in antibody tests (ZIKV should be considered too). It would be beneficial to provide the JEV-serology status of the study participants and/or their JEV vaccination status. Indeed, the authors discussed briefly potential cross-reactivity. However, the argument, that positivity in DENV PRNT indicates solely DENV reactivity is incorrect. Cross-reactivity among flaviviruses can also be seen in neutralization assays (especially after recent infection or vaccination). Therefore, testing against multiple flaviviruses and determining if the antibody titers differ significantly is the only option to draw definite conclusions. If the authors cannot provide these information, they should at least discuss the possibility of cross-reaction in more detail, especially in children (probably likely vaccinated against JEV).

The authors should provide more information on the serology results obtained with the various assays (especially the NS1, IgM/IgG RDT) as these are useful information to further characterize the study cohort. How many individuals where acute DENV-infected (vs how many convalescent), how many showed only IgM or IgM+IgG (primary vs secondary?)?

Even better performance makers than sensitivity and specificity are the positive and negative predictive values, especially when considering an assay for a specific application like pre-vaccination screening. It is great that the authors considered that. They could discuss these parameters when the DENV prevalence is higher then observed in this study. As mentioned by the authors, there seem to be regions in Thailand which higher prevalence. Would this affect their recommendation regarding the use of these assays for pre-vaccination screening?

Minor comments:

Line 54: the unit (NT50 or PRNT50) for the geometric mean antibody titer is missing. Also, be consistent by using only NT50 or PRNT50 describing titers and NT or PNRNT indicating the assay itself.

Be consistent on what WHO SAGE recommendation you refer to (reference no 4 from July 2016 used in introduction [line 75] or reference no 17 from September 2018 used in discussion [line272]). I'd recommend using the most recent one.

Line 110: correct to degree Celsius (℃)

Lines 144 and 256: Lübeck (umlaut diacritics can be typed by using first " followed by the vowel)

Line 163: indicate if you used different programs for the statistical analysis (SPSS) and the preparation of the plots/figures (looks like GraphPad)

Line 201: clarify what you mean by 'the ten thresholds'

Line 234: 'and' is doubled

Lines 248-249: correct sentence (e.g. 'shown by PRNT50')

Lines 249-250: correct sentence (e.g. 'was found to increase with age')

Lines 250-252: clarify why older individuals in this cohort were likely infected with DENV-1 before exposure to another DENV serotype. Reference to surveillance data would be helpful.

Line 253: 'Thai individuals' instead of 'Thais individuals'

Line 271: areas, regions or scenarios might be a better expression than 'scenes'

Line 278: clarify that you describe the assay sensitivity towards IgG (as the RDT can also measure IgM and NSI)

Lines 307-309: correct sentence as both parameters regard to rural settings which does not fit the argumentation

Lines 309-310: correct sentence (might delete or re-position 'with age')

Lines 327-329: clarify/simplify sentence to get your argument across

Reviewer #2: The idea is great. However, you have to design the experiment in a proper way to serve your goal. This manuscript might be considered for publication if you do all the analysis again in a proper manner. My main concerns are:

1- PRNT assay: You have to include proper virus strains in your experiment. There is data available for the strains you used, the passage number and the cells that you grew the virus. You must include titer for this experiment and also please add proper positive and negative control. Also, the way that you analyze the data for this experiment seems confusing. I added an article in the manuscript as an example of way that you can properly calculate the NT50. If possible, I would like to see the raw data for this assay.

2- The data analysis for ELISA assay also seems confusing. I would like to see the method that you used to obtain 20 relative unit/ml. Is it a standard number in the instruction of the kit?

3- In 3 assays, I could not be able to see the dilution factor for the serum samples. Are they the same for all the experiments?

4- Any strategy for discriminating false positive or false negative in the commercial kits? I know some labs are using some techniques like qPCR to verify the data?

5- Review of the discussion is pending and it must be done based on your new results and graphs.

I included some minor revisions in the manuscript but first of all, you must correct the analysis or clarify the present analysis to continue with your manuscript.

6. PLOS authors have the option to publish the peer review history of their article (what does this mean?). If published, this will include your full peer review and any attached files.

Reviewer #1: **Yes: **Heidi Auerswald

Reviewer #2: **Yes: **Touraj Aligholipour Farzani

---

## [Author Response · Author response to Decision Letter 0]

8 Jul 2021

PONE-D-21-09490

Dengue pre-vaccination screening test evaluation for the use of dengue vaccine in an endemic area

Reviewer #1 

Major issues:

1. A major part of flavivirus serology is unfortunately exclusion diagnostic. Therefore informative specificity analyses should include non-DENV samples, especially in a JEV endemic country as JEV is know to cause cross-reactivity in antibody tests (ZIKV should be considered too). It would be beneficial to provide the JEV-serology status of the study participants and/or their JEV vaccination status. Indeed, the authors discussed briefly potential cross-reactivity. However, the argument, that positivity in DENV PRNT indicates solely DENV reactivity is incorrect. Cross-reactivity among flaviviruses can also be seen in neutralization assays (especially after recent infection or vaccination). Therefore, testing against multiple flaviviruses and determining if the antibody titers differ significantly is the only option to draw definite conclusions. If the authors cannot provide these information, they should at least discuss the possibility of cross-reaction in more detail, especially in children (probably likely vaccinated against JEV).

Response Thank you for pointing this out. We agreed that cross-reactivity between flaviviruses had complicated the interpretations of results from serological assays. 

Dengue and Japanese encephalitis virus (JEV) are in the same flavivirus family which shared common antigenic epitopes and high cross-reactivities are not uncommon. In Thailand, a mouse brain-derived-inactivated JEV vaccine was initially included in the routine immunization schedule in 1990. Then, it became a nationwide vaccination in 2000 in Thai children at 12–18 months of age. For this reason, all participants in this study probably were vaccinated against JEV for at least nine years. However, the detailed vaccination history of all participants was not collected. Moreover, cases of JEV encephalitis have been reported, even in vaccinated cases, so natural JEV infection might not be ruled out. Convalescent anti-dengue antibody responses after dengue infection in JEV vaccinated and individuals who lived in JEV endemic areas were dengue-specific and higher than anti-JEV as determined by PRNT (Yamada, K., et al., Clin Diagn Lab Immunol 2003). A previous study showed that the sera of patients with JE showed cross-reactivity to DENV IgM and/or IgG ELISA, but the cross-reactivity was not detected in the neutralization tests against DENV. In addition, Saito Y. et al. reported cross-reactivity to DENV in an anti-DENV IgG ELISA after JEV vaccination. However, the DENV neutralizing antibodies were not detected in post-JEV immunization samples, indicating that neutralization tests are more specific than ELISA. Therefore, the serological data obtained from this study should be interpreted with caution since the samples were obtained in Thailand, where several known flaviviruses have been reported.

Nevertheless, in order to meet the reviewer's concern, we have performed additional PRNT data analysis using a higher cut-off. WHO recommended the use of NT90 instead of NT50 in endemic areas to decrease the background serum cross-neutralization among Flaviviruses. The result showed that the overall seropositive decreased from 81.7% by NT50 to 74.8% by NT80 and 68.7% by NT90. We also added these results in the S1 Table. The following sentence has also been added to the Discussion section, limitation part, page 19.

“Third, it is important to realize that cross-reactivity of anti-DENV antibodies with other flaviviruses could not be excluded by most commercial antibody-based assays. Dengue and Japanese encephalitis virus (JEV) are in the same flavivirus family, which shared common antigenic epitopes and high cross-reactivities are not uncommon. In Thailand, a mouse brain-derived-inactivated JEV vaccine was initially included in the routine immunization schedule in 1990. Then, it became a nationwide vaccination in 2000 in Thai children at 12–18 months of age [32]. For this reason, all participants in this study probably were vaccinated against JEV for at least nine years. However, the detailed vaccination history of all participants was not collected. Moreover, cases of JEV encephalitis have been reported even in vaccinated cases [33], so natural JEV infection might not be ruled out. Convalescent anti-dengue antibody responses after dengue infection in JEV vaccinated and individuals who lived in JEV endemic areas were dengue-specific and higher than anti-JEV as determined by PRNT [34]. A previous study showed that the sera of patients with JEV showed cross-reactivity to DENV IgM and/or IgG ELISA, but the cross-reactivity was not detected in the neutralization tests against DENV [35]. In addition, Saito Y. et al. reported cross-reactivity to DENV in an anti-DENV IgG ELISA after JEV vaccination. However, the DENV neutralizing antibodies were not detected in post-JEV immunization samples [36], indicating that neutralization tests are more specific than ELISA. Therefore, the serological data obtained from this study should be interpreted with caution since the samples were obtained in Thailand, where several known flaviviruses have been reported.”

S1 Table. Seroprevalence of DENV by PRNT (NT50, NT80, and NT90), IgG ELISA, and IgG rapid test.

Characteristics Seropositive

by NT50 Seropositive

by NT80 Seropositive

by NT90 Seropositive

by IgG ELISA Seropositive

by IgG rapid test

Overall 94 (81.7%) 86 (74.8%) 79 (68.7%) 82 (71.3%) 33 (28.7%)

Gender 

Male 58 (86.6%) 55 (82.1%) 50 (74.6%) 50 (74.6%) 20 (29.9%)

Female 36 (75.0%) 31 (64.6%) 29 (60.4%) 32 (66.7%) 13 (27.1%)

Age 

10-14 26 (68.4%) 22 (57.9%) 22 (57.9%) 21 (55.3%) 10 (26.3%)

15-18 32 (80.0%) 29 (72.5%) 26 (65.0%) 27 (67.5%) 8 (20.0%)

19-22 36 (97.3%) 35 (94.6%) 31 (83.8%) 34 (91.9%) 15 (40.5%)

2. The authors should provide more information on the serology results obtained with the various assays (especially the NS1, IgM/IgG RDT) as these are useful information to further characterize the study cohort. How many individuals where acute DENV-infected (vs how many convalescent), how many showed only IgM or IgM+IgG (primary vs secondary?)?

Response Thank you for the suggestion. We agree and added the serology results obtained from the RDT in the Results section, Seroprevalence of DENV amongst participants part, Page 12 as the following sentence.

“Based on the RDT, the result showed no positive NS1 result, indicating no early dengue infection case in this cohort. Moreover, there were 2 IgM positive cases, reflecting the asymptomatic acute primary dengue infection. There were 3 both IgM and IgG positive cases, suggesting a late primary or early secondary dengue infection.”

3. Even better performance makers than sensitivity and specificity are the positive and negative predictive values, especially when considering an assay for a specific application like pre-vaccination screening. It is great that the authors considered that. They could discuss these parameters when the DENV prevalence is higher then observed in this study. As mentioned by the authors, there seem to be regions in Thailand which higher prevalence. Would this affect their recommendation regarding the use of these assays for pre-vaccination screening?

Response Thank you for the suggestion. We agreed and added an improved discussion on these points to the Discussion section, page 17.

“In addition, the PPV for both tests was equal (100%), indicating that the probability that a person with a positive test will be truly seropositive was 100%. Therefore, both RDT and ELISA could be used to ensure who will be safe from the adverse effects of the vaccination. In contrast, the NPV was 25% and 63% for the RDT and ELISA. The probability that an individual with a negative RDT test will be truly seronegative was only 25%, and the rest of 75% may be seropositive. Therefore, if we use the RDT to identify PDI, those with a negative RDT test result will have a 75% chance of losing the opportunity from the vaccination.

Minor comments:

4. Line 54: the unit (NT50 or PRNT50) for the geometric mean antibody titer is missing. Also, be consistent by using only NT50 or PRNT50 describing titers and NT or PNRNT indicating the assay itself.

Response Thank you for your kind indication. We have added the unit (NT50) for the geometric mean antibody titer as suggested in this revision. 

“The highest anti-dengue antibody titers were recorded against DENV-1 and increased with age to a geometric mean NT50 titer (GMT) of 188.6 in the oldest age group.”

We also improved the consistency throughout the revised manuscript using only NT50 describing titers and PRNT indicating the assay itself as suggested.

5. Be consistent on what WHO SAGE recommendation you refer to (reference no 4 from July 2016 used in introduction [line 75] or reference no 17 from September 2018 used in discussion [line272]). I'd recommend using the most recent one.

Response Thank you for your suggestion. We agreed and replaced reference no 4 with the most recent WHO SAGE recommendation (reference no 17).

6. Line 110: correct to degree Celsius (℃)

Response Thank you for your kind indication. We have corrected the sentence as suggested.

7. Lines 144 and 256: Lübeck (umlaut diacritics can be typed by using first " followed by the vowel)

Response Thank you for your kind indication. We have corrected the word as suggested.

8. Line 163: indicate if you used different programs for the statistical analysis (SPSS) and the preparation of the plots/figures (looks like GraphPad)

Response We agreed and added the program used to prepare the figures in the Materials and methods section, Statistical Analysis part, Page 10 as the following sentence.

“figures were drawn using GraphPad Prism 8 (GraphPad Software Inc., California, USA).”

9. Line 201: clarify what you mean by 'the ten thresholds'

Response Thank you for the comment. We have clarified 'the ten thresholds' by adding 

“(NT50 titer<10)” at the end of the sentence.

“There were 31.6% naïve subjects in the 10-14 years old group, 20.0% of the 15-18 years old group and 2.7% of the 19-22 years old groups, and 18.3% of the overall sample had no detectable neutralizing dengue antibodies (NT50 titer<10, Table 2).”

10. Line 234: 'and' is doubled

Response Thank you for your kind indication. We have corrected the sentence as suggested.

“The relationship between NT50 titer and dengue IgG level (relative RU/mL by ELISA) was also examined.”

11. Lines 248-249: correct sentence (e.g. 'shown by PRNT50')

Response Thank you for your kind indication. We have corrected the sentence as suggested.

“The result indicated that approximately 82% of individuals neutralized one or more dengue serotypes shown by PRNT (NT50 titers ≥ 10).”

12. Lines 249-250: correct sentence (e.g. 'was found to increase with age')

Response Thank you for your kind indication. We have corrected the sentence as suggested.

“The proportion of individuals with exposure to more than one serotype was found to increase with age.”

13. Lines 250-252: clarify why older individuals in this cohort were likely infected with DENV-1 before exposure to another DENV serotype. Reference to surveillance data would be helpful.

Response We apologize for any confusion. We have revised the discussion to be more straightforward according to figure 2 and added references to surveillance data in the Discussion section, Page 15, as the following sentences.

“The predominant serotypes were DEN-1 and DEN-2 in the 10–14 and 15–18 age groups born during 2003-2011. Similar to the previous report, DENV-1 and DENV-2 were the most commonly reported serotypes in Thailand's national and/or regional studies between 2000 and 2010 (Limkittikul K. et al., PLoS Negl Trop Dis.. 2014). Besides, DEN-2 was predominant in the 19-22 years old age groups born during 1999-2002. This result was consistent with a previous surveillance study in the Ratchaburi province, which indicated that the predominant serotypes were DEN-2 in 1999 and 2002. (Anantapreecha S. et al., Dengue Bulletin. 2004)”

14. Line 253: 'Thai individuals' instead of 'Thais individuals'

Response Thank you for your kind indication. We have replaced “Thais individuals” with “Thai individuals” as suggested.

15. Line 271: areas, regions or scenarios might be a better expression than 'scenes'

Response Thank you for your kind indication. We have replaced “regions” with “scenes” as suggested.

16. Line 278: clarify that you describe the assay sensitivity towards IgG (as the RDT can also measure IgM and NSI)

Response Thank you. We agreed and revised the Discussion section, Page 17, as the following sentences.

“However, the RDT for detecting IgG demonstrated low sensitivity (35%), while the ELISA displayed greater sensitivity (87%).”

17. Lines 307-309: correct sentence as both parameters regard to rural settings which does not fit the argumentation

Response Thank you for your kind indication. We have corrected the sentence as suggested.

“Doum D. et al. reported that the IgG ELISA optical densities (OD) with age were consistently higher in rural settings (R2=0.46-0.54) and lower in urban settings (R2=0.15-0.28)”

18. Lines 309-310: correct sentence (might delete or re-position 'with age')

Response Thank you. We agreed and deleted “with age” as suggested.

“Our urban study indicated that the IgG ELISA titers (RU/ml) were low (R2=0.14).”

19. Lines 327-329: clarify/simplify sentence to get your argument across

Response Thank you. We agreed and revised the conclusion sentences as suggested.

“Due to their lower sensitivity of the RDT for identifying PDI, those with a negative RDT test result might lose the benefits of vaccination. Therefore, we do not recommend using the RDT to identify PDI in this specific setting. Anti-Dengue IgG ELISA is more suitable than the RTD for vaccine introduction and identifying which populations to target.” 

Reviewer #2: 

The idea is great. However, you have to design the experiment in a proper way to serve your goal. This manuscript might be considered for publication if you do all the analysis again in a proper manner. My main concerns are:

1. PRNT assay: You have to include proper virus strains in your experiment. There is data available for the strains you used, the passage number and the cells that you grew the virus. You must include titer for this experiment and also please add proper positive and negative control. Also, the way that you analyze the data for this experiment seems confusing. I added an article in the manuscript as an example of way that you can properly calculate the NT50. If possible, I would like to see the raw data for this assay.

Response Thank you for pointing this out. We have added more detail of cells and viruses and PRNT as suggested in the Materials and methods section, pages 5 and 7. We also added the raw data of the PRNT assay with this revision.

Cells and Viruses

The PRNT assays were completed in LLC-MK2 (Rhesus monkey kidney cells) obtained from ATCC. The Cells were grown in M-199 supplement with 20% heated-inactivated fetal bovine serum (HIFBS, Invitrogen, US), 1× L-glutamine (Invitrogen, US), 100 units of penicillin-streptomycin (P&S, Invitrogen, US) at 35˚C with 5% CO2 incubator. A concentration of cells at 4×104 per ml in M-199 with 10% HIFBS, 1× L-glutamine, 100 units of P&S was prepared and added 1.5 ml per well of 12-well plate, which continued cultured at 35˚C with 5% CO2 for 3 days before using in the PRNT assay. The four serotypes of DENV, including DENV-1 (16007), DENV-2 (16681), DENV-3 (16562), and DENV-4 (C0036/06), were used in the PRNT assay. Viruses were propagated in C6/36 mosquito cell line to obtain a sufficient titer (700PFU/ml) for testing. Virus titers were measure by plaque titration assay. Viruses were aliquot in small volume and stored at -70˚C until used without repeat freeze-thawing.

Plaque reduction neutralization test (PRNT)

PRNT was performed to determine the levels of virus type-specific neutralizing (NT) antibodies using standard methods, as previously described [7, 14, 15]. Briefly, the test serum was heat-inactivated at 56˚C for 30 minutes. In an ice bath, a 0.3 ml/tube of each 4-fold dilutions of the test sera beginning with a 1:10 dilution (1:10, 1:40, 1:160, 1:640 and 1:2,560) including a 0.3 ml serum diluent tube as a virus control well (MEM with 10%HIFBS) was mixed with an equal volume of each reference virus (DENV serotype noted above). Then the virus-serum mixture was incubated in a 35°C water bath for 1 hour. The virus-serum mixture was then inoculated (0 .1 ml per well) into the 3-day old LLC-MK2 cells in duplicate wells of a 12-well plate and incubated at room temperature for 1 hour on a rocker platform. After removing of excess virus-serum mixture, the first overlay medium, containing 0.9% low-melting point agarose (LMP Ultra PureTM LMP agarose, Invitrogen, USA), in house Hank’s BSS, 1X Vitamins (Invitrogen, US), 1X amino acids (Invitrogen, US), 5%HIFBS, L-glutamine, and 7.5% sodium bicarbonate, was added (1 ml per well) and allowed to solidify for 15 minutes. Plates were incubated at 35°C, 5% CO2 incubator for 4-6 days. At the end of the incubation period, stained cells with a second overlay medium (0.9% LMP, in house Hank’s BSS, 1X Vitamins, and 1X amino acids) containing 4% Neutral red (Sigma, US) 1 ml per well, and allowed to solidify for 15 minutes. Plates were incubated overnight at 35°C in a 5% CO2 incubator before counting the number of plaques by manual counting. Positive control wells (virus without sera) were established for each assay run to ensure infectivity of the cell monolayer (formation of 25- 50 plaques/well was considered acceptable). Positive serum control was pooled high-titered convalescent sera against four dengue serotypes diluted with pooled normal human sera to get appropriate NT50 endpoint and aliquot in small volume and frozen storage for use. Plaque reduction neutralization endpoints were calculated using probit analysis. Briefly, the percentage of plaques counted in test sera was compared with the number of plaques from the control preparation. Log dilution of test sera preparation (i.e., 1:10, 1:40, 1:160, 1:640, 1:2,560) were plotted along the X-axis, whereas percent reduction in plaque count was detailed along the Y-axis. Percent count plaque reductions for each test sample at each dilution was plotted, and a best-fit line was drawn. The reciprocal of the lowest dilution of test sera to neutralize 50% of the control virus input represents the NT50 titer. Therefore, the NT50 titer is calculated by counting plaques and reporting the titer as the reciprocal of the last serum dilution to show a 50% reduction of the control plaque count based on the back-titration of control plaques. In this study, NT50 titers≥10 were considered positive for virus-neutralizing activity. An individual who has antibody titers <10 for the four serotypes was classified as naïve. An individual who has NT50 antibody titers ≥10 to only one serotype (no NT titer to other serotypes) was classified as monotypic. Correspondingly, an individual who has NT50 antibody titers ≥10 for more than one serotype was classified as multitypic.

2. The data analysis for ELISA assay also seems confusing. I would like to see the method that you used to obtain 20 relative unit/ml. Is it a standard number in the instruction of the kit?

Response We apologize for any confusion. The IgG results were reported quantitatively according to the manufacturer’s instructions described as follows.

Photometric measurement of the color intensity should be made at a wavelength of 450 nm and a reference wavelength between 620 nm and 650 nm within 30 minutes of adding the stop solution. The standard curve from which the concentration of antibodies in the patient samples can be taken is obtained by point-to-point plotting of the extinction values measured for the 3 calibration sera against the corresponding units (linear/linear). Use “point-to-point” plotting for calculation of the standard curve by computer. The following plot is an example of a typical calibration curve (from the manufacturer’s instructions).

The upper limit of the normal range of non-infected persons (Cut-off value) recommended by EUROIMMUN is 20 relative units (RU)/ml. EUROIMMUN recommends interpreting results as follows: <16 RU/ml: negative, ≥16 to <22 RU/ml: borderline, and ≥22 RU/ml: positive.

In this revision, we have added more detail on the data analysis for the ELISA assay in the Materials and methods section, Dengue IgG ELISA part, Page 9-10.

“Photometric measurement of the color intensity was made at a wavelength of 450 nm and a reference wavelength between 620 nm and 650 nm within 30 minutes of adding the stop solution. The IgG results were reported quantitatively according to the manufacturer’s instructions using the standard curve obtained by point-to-point plotting of the extinction values measured for the three calibration sera against the corresponding units (linear/linear). A specimen was considered IgG positive if the result was ≥22 relative units (RU)/ml, borderline if ≥16 and <22 RU/ml, and negative if <16 RU/ml.”

3. In 3 assays, I could not be able to see the dilution factor for the serum samples. Are they the same for all the experiments?

Response We apologize that we were not clear about the dilution factor in the first submission.

For PRNT, serum was diluted in a four-fold serial dilution starting from 1:10 to 1:2,560 as already mentioned in the Materials and methods section, Plaque Reduction Neutralization Test (PRNT) part, Page 7.

For RDT, undiluted serum samples were used for detecting NS1, IgG, and IgM. We have added this point in the Materials and methods section, Dengue duo rapid test kit part, Page 8.

“For the NS1 antigen, 100 μL of undiluted serum sample was added to the left sample well, and the results were read within 15-20 minutes. For IgG/IgM, 10 μL of undiluted serum sample was added to the right sample well, and four drops (90-120 μL) of assay diluent were added to the round assay diluent well.”

For the dengue IgG ELISA, serum samples were diluted at 1:101, as already mentioned in the Materials and methods section, Dengue IgG ELISA part, Page 9.

4. Any strategy for discriminating false positive or false negative in the commercial kits? I know some labs are using some techniques like qPCR to verify the data?

Response This study aimed to conduct a serosurvey for past dengue infections to support decision-making for vaccine introduction and identify which populations to target.

Nucleic acid detection like qPCR can be used to diagnose the infection during the early stages of the disease. At the end of the acute phase of infection, serology is the method of choice for diagnosis.

The PRNT is currently considered to be the "gold standard" to characterize and quantify circulating levels of anti-DENV neutralizing antibodies. For this reason, the PRNT was used as the reference test in this study for evaluating the performance of commercial kits in identifying PDI.

We have performed additional data analysis using PRNT as the reference test. The result showed no false-positive rate (0%) for both ELISA and RDT. However, there was a 12.8% false-negative rate for the ELISA and a 64.9% false-negative rate for the RDT in identifying PDI. 

5. Review of the discussion is pending and it must be done based on your new results and graphs. I included some minor revisions in the manuscript but first of all, you must correct the analysis or clarify the present analysis to continue with your manuscript.

Response We have added the suggested content to the manuscript on where the change can be found as a yellow highlight in the revised manuscript.

6. You must include the criteria here, how do you considered them healthy?

Response We have added healthy children's definition in the Materials and methods section, Study Location and Population part, page 6 as the following sentences.

“Healthy children were defined as participants in good health based on medical history and physical examination. We excluded children with acute febrile illness (body temperature more than 37.5˚C) or moderate or severe acute illness/infection on the day of enrolment, according to the investigator's judgment.”

7. Any specific reason for these groups?????? Any specific reason for these geographical area?? Is there any available previous data from these regions???

Response Ratchaburi province is located approximately 100 km west of Bangkok. It lies between the Maeklong River on the east and the Thai-Myanmar border on the west. We have a collaborative research network in this province. A previous dengue disease surveillance study on approximately 3000 primary-school children in seven schools in Muang district of Ratchaburi province indicated that approximately four percent of the children had laboratory-confirmed dengue per year, and all four dengue virus types were found to be the causes of illness in children in all seven schools. Moreover, the study pointed out that the Muang district of Ratchaburi province could be suitable for dengue vaccine testing (Sabchareon A. et al., PLoS Negl Trop Dis. 2012). We have added this point in the Discussion section, page 15.

8. For Table 2, it will be grate to add graph for this data 

Response We have changed the data in Table 2 to Figure 1 as suggested.

---

## [Decision Letter · Decision Letter 1]

27 Jul 2021

PONE-D-21-09490R1

Dengue pre-vaccination screening test evaluation for the use of dengue vaccine in an endemic area

PLOS ONE

Dear Dr. Srisawat,

Thank you for submitting your manuscript to PLOS ONE. After careful consideration, we feel that it has merit but does not fully meet PLOS ONE’s publication criteria as it currently stands. Therefore, we invite you to submit a revised version of the manuscript that addresses the last points raised below during the review process.

We look forward to receiving your revised manuscript.

Kind regards,

Ray Borrow, Ph.D., FRCPath

Academic Editor

PLOS ONE

Journal Requirements:

Reviewers' comments:

Reviewer's Responses to Questions

**Comments to the Author**

1. If the authors have adequately addressed your comments raised in a previous round of review and you feel that this manuscript is now acceptable for publication, you may indicate that here to bypass the “Comments to the Author” section, enter your conflict of interest statement in the “Confidential to Editor” section, and submit your "Accept" recommendation.

Reviewer #1: All comments have been addressed

Reviewer #2: All comments have been addressed

2. Is the manuscript technically sound, and do the data support the conclusions?

Reviewer #1: Yes

Reviewer #2: Yes

3. Has the statistical analysis been performed appropriately and rigorously? 

Reviewer #1: Yes

Reviewer #2: Yes

4. Have the authors made all data underlying the findings in their manuscript fully available?

Reviewer #1: Yes

Reviewer #2: Yes

5. Is the manuscript presented in an intelligible fashion and written in standard English?

Reviewer #1: Yes

Reviewer #2: No

6. Review Comments to the Author

Reviewer #1: (No Response)

Reviewer #2: Thank you very much for your precise responses. You addressed almost all my concerns. After some minor revisions, this manuscript is ready to be published.

1- I added some minor recommendations in the manuscript.

2- Please send the manuscript for English editing.

3- You can make the discussion more better.

7. PLOS authors have the option to publish the peer review history of their article (what does this mean?). If published, this will include your full peer review and any attached files.

Reviewer #1: **Yes: **Heidi Auerswald

Reviewer #2: No

---

## [Author Response · Author response to Decision Letter 1]

25 Aug 2021

PONE-D-21-09490

Dengue pre-vaccination screening test evaluation for the use of dengue vaccine in an endemic area

Journal Requirements: Please review your reference list to ensure that it is complete and correct. If you have cited papers that have been retracted, please include the rationale for doing so in the manuscript text, or remove these references and replace them with relevant current references. Any changes to the reference list should be mentioned in the rebuttal letter that accompanies your revised manuscript. If you need to cite a retracted article, indicate the article’s retracted status in the References list and also include a citation and full reference for the retraction notice.

Response We have reviewed and revised references as suggested. We confirmed that no retracted article was cited in this manuscript. We removed the references “Doum D, Overgaard HJ, Mayxay M, Suttiprapa S, Saichua P, Ekalaksananan T, et al. Correction: Doum, D., et al. Dengue Seroprevalence and Seroconversion in Urban and Rural Populations in Northeastern Thailand and Southern Laos. Int J Environ Res Public Health. 2021;18(4).” and replaced them with “Doum D, Overgaard HJ, Mayxay M, Suttiprapa S, Saichua P, Ekalaksananan T, et al. Dengue Seroprevalence and Seroconversion in Urban and Rural Populations in Northeastern Thailand and Southern Laos. Int J Environ Res Public Health. 2020;17(23).” Moreover, following references were added in this revision as the reviewer's suggestion.

• Ref 3. Hadinegoro SR, Arredondo-García JL, Capeding MR, Deseda C, Chotpitayasunondh T, Dietze R, et al. Efficacy and Long-Term Safety of a Dengue Vaccine in Regions of Endemic Disease. N Engl J Med. 2015;373(13):1195-206. 

• Ref 4. Wilder-Smith A, Vannice KS, Hombach J, Farrar J, Nolan T. Population Perspectives and World Health Organization Recommendations for CYD-TDV Dengue Vaccine. J Infect Dis. 2016;214(12):1796-9. 

• Ref 5. Sridhar S, Luedtke A, Langevin E, Zhu M, Bonaparte M, Machabert T, et al. Effect of Dengue Serostatus on Dengue Vaccine Safety and Efficacy. N Engl J Med. 2018;379(4):327-40.

• Ref 10. Roehrig JT, Hombach J, Barrett AD. Guidelines for Plaque-Reduction Neutralization Testing of Human Antibodies to Dengue Viruses. Viral Immunol. 2008;21(2):123-32. 

• Ref 11. Raafat N, Blacksell SD, Maude RJ. A review of dengue diagnostics and implications for surveillance and control. Trans R Soc Trop Med Hyg. 2019;113(11):653-60.

• Ref 13. Luo R, Fongwen N, Kelly-Cirino C, Harris E, Wilder-Smith A, Peeling RW. Rapid diagnostic tests for determining dengue serostatus: a systematic review and key informant interviews. Clin Microbiol Infect. 2019;25(6):659-66.

• Ref 26. Wilder-Smith A, Smith PG, Luo R, Kelly-Cirino C, Curry D, Larson H, et al. Pre-vaccination screening strategies for the use of the CYD-TDV dengue vaccine: A meeting report. Vaccine. 2019;37(36):5137-46.

Reviewer #2: I added some minor recommendations in the manuscript.

1. It is appropriate to mention that virus is circulating in Thailand. You analyzed the people in a provenance.

Response Thank you for the suggestion. We have revised the conclusion part as the following sentence.

“Almost 80% of adolescents and youth in Ratchaburi province had already been exposed to one or more of the dengue virus serotypes. The dengue IgG RDT displayed low sensitivity and is likely not be suitable for dengue pre-vaccination screening. These results support the use of IgG ELISA test for dengue vaccination in endemic areas.”

2. Please add some references in the introduction

Response Thank you for the comment. We have added references in the introduction section, page 4-5 as suggested.

“The efficacy trials showed that Dengvaxia protected against severe dengue in people who had exposure to dengue prior to vaccination.[3-5]”

“However, PRNT is a labor-intensive, relatively costly test making it inconvenient for routine surveillance purposes.[10, 11]”

“A recent systematic review of RDT showed sensitivities and specificities between 80% and 100% for dengue IgG detection compared to ELISA.[13]”

• Ref 3. Hadinegoro SR, Arredondo-García JL, Capeding MR, Deseda C, Chotpitayasunondh T, Dietze R, et al. Efficacy and Long-Term Safety of a Dengue Vaccine in Regions of Endemic Disease. N Engl J Med. 2015;373(13):1195-206. 

• Ref 4. Wilder-Smith A, Vannice KS, Hombach J, Farrar J, Nolan T. Population Perspectives and World Health Organization Recommendations for CYD-TDV Dengue Vaccine. J Infect Dis. 2016;214(12):1796-9. 

• Ref 5. Sridhar S, Luedtke A, Langevin E, Zhu M, Bonaparte M, Machabert T, et al. Effect of Dengue Serostatus on Dengue Vaccine Safety and Efficacy. N Engl J Med. 2018;379(4):327-40.

• Ref 10. Roehrig JT, Hombach J, Barrett AD. Guidelines for Plaque-Reduction Neutralization Testing of Human Antibodies to Dengue Viruses. Viral Immunol. 2008;21(2):123-32. 

• Ref 11. Raafat N, Blacksell SD, Maude RJ. A review of dengue diagnostics and implications for surveillance and control. Trans R Soc Trop Med Hyg. 2019;113(11):653-60.

• Ref 13. Luo R, Fongwen N, Kelly-Cirino C, Harris E, Wilder-Smith A, Peeling RW. Rapid diagnostic tests for determining dengue serostatus: a systematic review and key informant interviews. Clin Microbiol Infect. 2019;25(6):659-66.

3. Do you have the passage number of the viruses? It is important to add the passage number

Response We have added the passage number of the viruses in the Materials and methods section, Cells and Viruses part, page 5-6 as suggested.

“The four serotypes of DENV including DENV-1 (16007-Passage SM-2 C6/36-5), DENV-2 (16681-Passage MIK2-3, C6/36-1, SM-2, C6/3), DENV-3 (16562-Passage MK2-3, C6/36-2, SM-1, C6/36-6) and DENV-4 (C0036/06-Passage C6/36-8) were used in the PRNT assay.”

4. You have to titer the virus after first freeze.

Response We have added this point in the Materials and methods section, Cells and Viruses part, page 6 as suggested.

“We titered the virus after the first freeze using a procedure similar to PRNT assay to determine virus titer and also for virus dilution to reach 70 PFU/100 ul.”

5. We observed the plates daily for the onset of CPEs?

Response CPEs observation has been done daily only during the step of virus propagation to ensure that we harvested virus at the right time. However, for PRNT assay we don’t do that as the appropriate incubation period has already been set up for a particular DENV serotype to get the suitable plaque size for counting.

6. Please send the manuscript for English editing.

Response The grammar was rechecked and corrected by the medical manuscript editor from Research Affairs, Faculty of Medicine Chulalongkorn University (Michael D. Ullman, Ph.D.).

7. You can make the discussion more better.

Response We very much appreciate reviewer's suggestion and we have revised the Discussion section by adding some sentences (See page 17 and 19) as follows.

“According to WHO recommendations [7], the screening tests should have a high specificity to minimize individual risk and the inadvertent use of the vaccine in seronegative persons who have false-positive test results. The test should also have high sensitivity to maximize individual and population benefit by identifying a high proportion of previously exposed persons who can benefit from vaccination. The Global Dengue and Aedes-transmitted diseases Consortium (GDAC) suggested that any decision about implementing a prevaccination screening strategy with the commercially available tests at the country level requires cautious evaluation. In a high seroprevalence area, a test with very high sensitivity is needed. In a low seroprevalence area, a very high specificity test is crucial; however, a national implementation may not be cost-effective.[26]”

“In this study, we used the ELISA index cutoff values for seropositivity as stated in the product insert. However, it was unclear whether the ELISA cutoff values recommended by the manufacturer were suitable for prevaccination screening.”

---

## [Editor Report · Decision Letter 2]

26 Aug 2021

Dengue pre-vaccination screening test evaluation for the use of dengue vaccine in an endemic area

PONE-D-21-09490R2

Dear Dr. Srisawat,

We’re pleased to inform you that your manuscript has been judged scientifically suitable for publication and will be formally accepted for publication once it meets all outstanding technical requirements.

Kind regards,

Ray Borrow, Ph.D., FRCPath

Academic Editor

PLOS ONE
---

## [Editor Report · Acceptance letter]

31 Aug 2021

PONE-D-21-09490R2 

Dengue pre-vaccination screening test evaluation for the use of dengue vaccine in an endemic area 

Dear Dr. Srisawat:

I'm pleased to inform you that your manuscript has been deemed suitable for publication in PLOS ONE. Congratulations! Your manuscript is now with our production department. 

Kind regards, 

on behalf of

Prof. Ray Borrow 

Academic Editor

PLOS ONE